# Motion Smoothness-Based Assessment of Surgical Expertise: The Importance of Selecting Proper Metrics

**DOI:** 10.3390/s23063146

**Published:** 2023-03-15

**Authors:** Farzad Aghazadeh, Bin Zheng, Mahdi Tavakoli, Hossein Rouhani

**Affiliations:** 1Department of Mechanical Engineering, University of Alberta, Edmonton, AB T6G 1H9, Canada; farzad@ualberta.ca; 2Department of Surgery, University of Alberta, Edmonton, AB T6G 2B7, Canada; 3Department of Electrical and Computer Engineering, University of Alberta, Edmonton, AB T6G 2R3, Canada

**Keywords:** movement smoothness, motion jerk, surgical skill assessment, motion tracking, minimally invasive surgery

## Abstract

The smooth movement of hand/surgical instruments is considered an indicator of skilled, coordinated surgical performance. Jerky surgical instrument movements or hand tremors can cause unwanted damages to the surgical site. Different methods have been used in previous studies for assessing motion smoothness, causing conflicting results regarding the comparison among surgical skill levels. We recruited four attending surgeons, five surgical residents, and nine novices. The participants conducted three simulated laparoscopic tasks, including peg transfer, bimanual peg transfer, and rubber band translocation. Tooltip motion smoothness was computed using the mean tooltip motion jerk, logarithmic dimensionless tooltip motion jerk, and 95% tooltip motion frequency (originally proposed in this study) to evaluate their capability of surgical skill level differentiation. The results revealed that logarithmic dimensionless motion jerk and 95% motion frequency were capable of distinguishing skill levels, indicated by smoother tooltip movements observed in high compared to low skill levels. Contrarily, mean motion jerk was not able to distinguish the skill levels. Additionally, 95% motion frequency was less affected by the measurement noise since it did not require the calculation of motion jerk, and 95% motion frequency and logarithmic dimensionless motion jerk yielded a better motion smoothness assessment outcome in distinguishing skill levels than mean motion jerk.

## 1. Introduction

Surgical complications threaten patients’ safety and impose significant costs on the healthcare system. It was reported that the frequency rate of surgical complications is 8–12% worldwide [1]. In the United States alone, surgical complications resulted in 32,600 deaths, 2.4 million extra days in hospitals, and USD 9.3B of costs in a year [2]. Therefore, it is essential to ensure that surgeons and surgical residents acquire sufficient surgical skills prior to performing surgeries in operating rooms because unskilled surgical performance is associated with increased surgical complications [3]. Differing from the assessment of medical knowledge via written/oral examinations, the assessment of technical hands-on surgical skills can be more challenging [4].

Currently, the assessment of technical surgical skills is often carried out by senior surgeons using checklists such as OSATS (Objective Structured Assessment of Technical Skills) [5] for open surgeries, GOALS (Global Operative Assessment of Laparoscopic Skills) [6] for laparoscopic surgeries, and GEARS (Global Evaluative Assessment of Robotic Skills) [7] for robotic surgeries. Specifically, senior surgeons observe the performance of surgical residents/junior surgeons during operations or post-operatively by watching recorded videos. However, this approach is subjective, bias-prone, and labour-intensive. 

To tackle the shortcomings associated with surgical skill assessments using checklists, motion analysis of surgical instruments/hands has been implemented to assess surgical skill levels quantitatively and objectively. The motion data recorded from surgical instruments and hands can characterize skill-related factors such as depth perception and bimanual dexterity, which have been included in checklists [6]. Various motion tracking approaches, such as electromagnetic motion tracking [8,9], image processing [10,11], and video recordings [12,13], have been implemented to assess the motion of surgical instruments/hands. 

While interpreting the motion data of surgical instruments/hands, motion smoothness is one of the metrics that has been used to classify surgical skill levels. It was asserted that smooth movements can indicate a skilled performance [14,15]. Although surgical instrument path length and task duration can reveal significant differences among surgical skill levels [10,12,16], they cannot reveal all the aspects of surgical expertise. In real-world surgical tasks, surgical trainees may finish the tasks with a relatively low path length and task duration, but they may commit detrimental mistakes due to the jerky movements of surgical instruments or hand tremors. Jerky movements or hand tremors can affect surgical outcomes and can cause damage to the delicate tissues of a surgical site, especially in delicate surgical procedures such as microsurgery [17,18,19]. Hence, elements such as respect for tissue and instrument handling, which are related to motion smoothness, are included in the surgical skill assessment checklists, e.g., OSATS [5] and GOALS [6]. Most expert surgeons move their hands and surgical instruments smoothly without sudden changes in movement acceleration, whereas novice surgeons are not able to regulate their hand movements due to insufficient movement control. Therefore, the assessment of motion smoothness is a feasible way to distinguish between expert and novice surgeons. Various metrics were used for motion smoothness assessment [10,16,20,21,22,23,24], which caused some extent of inconsistency in the reported results. 

Chmarra et al. [16] reported the time-integrated squared motion jerk of surgical instruments for simulated laparoscopic tasks and showed that gynecologists and gynecologic residents moved surgical instruments more smoothly than medical interns. Nevertheless, this finding was only observed from the dominant hand but not from the non-dominant hand. Using the mean motion jerk of surgical instruments, Davids et al. [20] found that expert surgeons had lower mean motion jerk than novice surgeons during a neurosurgery dissection task. However, Maithel et al. [21] did not report significant differences between junior and senior residents using the same metric (mean motion jerk) to calculate motion smoothness. They implemented the mean motion jerk of surgical instruments during a triangular item transfer task in a simulated laparoscopic surgery setting. Sanchez-Margallo et al. [22] used the mean motion jerk of surgical instruments to evaluate the instrument motion smoothness between intermediate and expert surgeons in simulated laparoscopic tasks such as synthetic fabric cutting, organic tissue dissection, and organic tissue suturing tasks. They found that the expert surgeons had significantly lower mean instrument motion jerk values of the dominant hand compared to intermediate surgeons in the synthetic fabric cutting task. Mansoor et al. [23] found that expert surgeons compared to novice surgeons had a higher mean motion jerk of the instrument held in the left hand during suture and tie along with tube ligation tasks, whereas novice surgeons had higher mean motion jerk of both instruments in a precision cutting task using a laparoscopic simulator. 

Dimensionless motion jerk, instead of motion jerk, has been used in other studies as a metric to analyse instrument motion smoothness. Using this metric, Ghasemloonia et al. [24] revealed significant differences between surgeons–surgical residents and gamers–engineers’ groups in terms of surgical instrument motion smoothness during simulated tasks, i.e., peg-in-hole tasks. They stated that surgeons–surgical residents had smoother instrument movements than gamers–engineers. However, when Oropesa et al. [10] used this metric, they could not find that motion smoothness significantly differed across expert surgeons, surgical residents, and novices during a peg transfer task in a simulated laparoscopic setting. This may have resulted from the formula used by them as they used the magnitude of the position vector, which is discussed in the discussion section in detail. 

We believe that the conflicting results in previous studies regarding motion smoothness assessment among surgical skill levels were due to the different metrics used for calculating motion smoothness. Hence, we aimed to implement several motion smoothness metrics in our study and evaluate their outcomes for surgical skill assessment. Particularly, we introduced a novel metric called 95% frequency of surgical tooltip motion. Previous metrics calculating motion smoothness in the time domain rely on motion jerk calculation, which is inherently affected by differentiation noise. Our method reported motion smoothness in the frequency domain, which is more robust to the data noise. The metrics implemented in the present study were: (i) mean motion jerk of surgical tooltips, (ii) logarithmic dimensionless motion jerk of surgical tooltips, and (iii) 95% frequency of surgical tooltips motion. We hypothesized that higher surgical skill levels, such as expert surgeons, have higher motion smoothness compared to lower surgical skill levels, such as novices. 

## 2. Materials and Methods

### 2.1. Research Environment

A control laboratory study was conducted in the Surgical Simulation Research Lab of the University of Alberta. We used a laparoscopic surgical tower (Stryker Corporation, Kalamazoo, MI, USA) along with a laparoscope (Stryker Corporation, Kalamazoo, MI, USA) to capture the simulated surgical site view (Figure 1a). We placed a surgical task board inside a box (30.5 × 30.5 × 21.5 cm) to simulate a laparoscopic surgical setting (Figure 1a). 

### 2.2. Participants

Four attending general surgeons as the expert group (1177 ± 355 minimally invasive surgery cases), five surgical residents as the intermediate group (26 ± 13 minimally invasive surgery cases), and nine participants without minimally invasive surgery cases as the novice group took part in the study. Informed consent was obtained from each participant before entering the study. The study protocol was approved by the Research Ethics Review Board of the University of Alberta (Study ID: Pro00114163). 

### 2.3. Tasks 

Prior to data collection, the author (FA) demonstrated the tasks to participants, and they were asked to practise each task once to get acquainted (self-assessed) with the tasks. We included three tasks in this study:

(1)Peg transfer task: The participants grasped a peg (1.6 × 1.4 × 1.6 cm) from a pin on the task board inside a box (Figure 1a) using the laparoscopic grasper (Ethicon Endo-Surgery, Cincinnati, OH, USA) held in the non-dominant hand. After that, they delivered the peg to the grasper held in the dominant hand, moved it to the target (a pin 10 cm away from the initial pin), and dropped it there. Then, the same procedure was conducted for the second peg. Once the two pegs were placed on the dominant hand’s side, the participants conducted the reverse procedure in order to transfer the pegs back to the original position. The peg transfer task is included in the Fundamentals of Laparoscopic Surgery (FLS) curriculum [25]. (2)Bimanual peg transfer task: Using both surgical graspers held in the dominant and non-dominant hands, the participants grasped the two pegs simultaneously from the non-dominant hand’s side, moved them to the targets (pins 10 cm away from the initial pins’ position), and dropped them there. Afterwards, participants grasped the pegs from the dominant hand’s side and moved them back to the initial positions. This task required a high level of bimanual coordination.(3)Rubber band translocation task: A rubber band was placed around four pins on the task board, and participants grasped the rubber band with two laparoscopic graspers, translocated the rubber band to the distal pins (5.5 cm away from the initial pins), released the rubber band, re-grasped it, and moved it back to the original position. This task was included as it had tool–tissue interaction that could affect surgical performance. This task also required a high level of bimanual coordination.

### 2.4. Motion Tracking of Surgical Tooltip

To acquire the position data of surgical instruments, we used the OptiTrack Flex 13 (NaturalPoint, Inc., Corvallis, OR, USA) motion capture system with a sampling frequency of 120 Hz, accuracy and precision of 0.2 mm, based on the manufacturer’s provided information and our preliminary observations. Afterwards, the high-frequency noise in the recorded position data was removed by low-pass filtering the position data using a zero-lag fourth-order Butterworth filter with a cut-off frequency of 5 Hz.

The surgical tooltips were placed inside the training box, not visible to motion capture cameras (Figure 1a). We could not place markers on them to track their position. Therefore, we placed three markers on the front side (Figure 1b) and an additional marker on the back side of the tool handles. Moreover, prior to actual data collection, the configuration of the motion capture system was adjusted to minimize marker occlusions. Placing an additional marker and adjusting the motion capture configuration ensured that at least three markers were visible during the actual trials. Hence, we were able to compute the relative position of the surgical tooltips with respect to the tool handle markers. Compared to other motion capture approaches, our proposed approach has the potential to be implemented in real-world surgeries since it is not sensitive to laparoscope movements and light changes, which frequently take place in real-world surgeries. In comparison, other methods, such as image processing algorithms and video recordings used for computing surgical tooltip position, are often interrupted by laparoscope movements and light changes. Electromagnetic motion tracking systems are affected by interference with other metallic instruments, such as laparoscopes, which is critical in any laparoscopic surgery. 

### 2.5. Motion Smoothness Derivation Algorithms

Having derived the right and left tooltips’ positions, we were able to derive the tooltip motion smoothness metrics (Table 1). Three tooltip motion smoothness algorithms were used in this study: (1)Mean tooltip motion jerk (J) was defined as the mean value of the magnitude of the tooltip motion jerk (the third time derivative of the tooltip position).(2)Logarithmic dimensionless tooltip motion jerk (DJ) was derived from tooltip motion jerk converted into a dimensionless metric via normalizing by tooltip path length (PL) and task duration (T). The term $\int_{t=0}^{T}{\left|\frac{d^{3}\vec{r}}{dt^{3}}\right|}^{2}dt$ in Table 1 has a dimension of $PL^{2}.T^{-5}$; hence, it is normalized using $\frac{T^{5}}{PL^{2}}$. This normalization is in accordance with [10,14,24]. Since dimensionless motion jerk values had different orders of magnitude across the skill levels, logarithmic values were used to report the results. To derive this metric, we required the tooltip path length, computed by integrating the tooltip velocity magnitude with respect to time. (3)The 95% tooltip motion frequency (f_95%_) was calculated in the frequency domain, as opposed to the first two metrics above that were calculated in the time domain. To calculate f_95%_, we converted the tooltip position data into the frequency domain using the power spectral density of the tooltip position. The power spectral density of the tooltip position was calculated by the pwelch function (10 s Hamming windows and 50% overlap) of MATLAB 2020, implementing Welch’s method. Then, we identified the frequency below which contained 95% of the total power of the tooltip position (Figure 2). We intended not to include the high-frequency and low-amplitude content of the tooltip position signal in the motion smoothness metric calculation, which resulted from the measurement noise. Therefore, we considered 95% of the total power of the tooltip position to derive the motion smoothness metric, neglecting the area under the power spectral density generated from the measurement noise (5%).

### 2.6. Statistical Analysis

A Shapiro–Wilk test indicated that the motion smoothness metrics were not normally distributed (*p* < 0.05). Therefore, we used the non-parametric test of Kruskal–Wallis to investigate the significant differences in tooltip motion smoothness among the three surgical skill levels. The significant differences between each pair of the two surgical skill levels were perused using the Mann–Whitney U test. The significance level was chosen at *p* < 0.05, and the statistical analysis was conducted via SPSS v. 26 (SPSS Inc., Chicago, IL, USA). The *p*-values associated with the Kruskal–Wallis and Mann–Whitney U tests are shown in Table 2.

## 3. Results

### 3.1. Mean Tooltip Motion Jerk

Mean tooltip motion jerk (J) did not discriminate among the three surgical skill levels in terms of tooltip motion smoothness except for the instrument held in the non-dominant hand during the peg transfer task (Figure 3a,b). Surprisingly, studying pairwise comparisons, J represented higher values for experts compared to novices in the peg transfer task and for the instrument in the non-dominant hand. In other words, this indicated that experts compared to novices had jerkier tooltip movements as opposed to our hypothesis that experts perform smoother tooltip movements.

### 3.2. Logarithmic Dimensionless Tooltip Motion Jerk

Logarithmic dimensionless tooltip motion jerk (DJ) differed significantly across the surgical skill levels for both instruments and in all the tasks (Figure 3c,d). DJ showed a lower value in the higher surgical skill level. Pairwise comparisons revealed that experts had a lower DJ value than novices for both instruments and in all the tasks. Additionally, a lower DJ value was observed for experts compared to intermediates in the peg and bimanual peg transfer tasks for both instruments. Also, intermediates showed a lower DJ value compared to novices in the peg and bimanual peg transfer tasks for both instruments.

### 3.3. The 95% Tooltip Motion Frequency

The 95% tooltip motion frequency (f_95%_) represented significant differences across the three skill levels for both instruments and in all the tasks. A decreasing trend of f_95%_ was observed from lower surgical skill levels to higher ones (Figure 3e,f). According to pairwise comparisons, experts had lower f_95%_ values than novices in all the task-instrument conditions. Intermediates revealed lower f_95%_ values than novices in the peg transfer task for both instruments and in the rubber band translocation task for the instrument held in the non-dominant hand.

### 3.4. Tooltip Path Length and Task Duration

Since tooltip path length (PL) and task duration (T) values were employed to derive DJ values from tooltip motion jerk, PL and T can play an important role in understanding the difference between the J and DJ values. Therefore, we compared PL and T across the three surgical skill levels (Figure 4). PL for both instruments and T differed significantly across the surgical skill levels in all the tasks. Perusing pairwise comparisons, we observed that experts accomplished all the tasks with a significantly shorter PL of both instruments and a significantly shorter T compared to novices. Additionally, intermediates compared to novices, had a shorter PL of both instruments in the bimanual peg transfer task and a shorter PL of the instrument held in the non-dominant hand in the rubber band translocation task. Intermediates also performed the peg transfer and bimanual peg transfer tasks with a significantly shorter T than novices. Experts, compared to intermediates, conducted the peg transfer and bimanual peg transfer tasks with a significantly shorter T and had a shorter PL for the instrument held in the dominant hand. Moreover, the sample tooltip trajectories, tooltip PL and T for the peg transfer, bimanual peg transfer, and rubber band translocation tasks are shown in Figure 5, Figure 6 and Figure 7, respectively.

## 4. Discussion

In this study, we used three algorithms to compute the motion smoothness by taking data from tooltips in a simulated laparoscopic surgery setting. We evaluated the motion smoothness algorithms for distinguishing three surgical skill levels.

We found that mean tooltip motion jerk (J) was not capable of discrimination between surgical skill levels; therefore, it was not a responsive metric for motion smoothness assessment. This conflicting observation originated from how J was computed. J was derived based on the third time derivative of the tooltip position, which is a function of task duration and path length [14]. In other words, J would have different values if the same level of motion smoothness was maintained during the trials of the same task with different trial durations, e.g., with different rest periods. In this case, the trial with the longer duration represents lower J values, whereas both trials should have the same level of motion smoothness. Path length may be another element that needs to be considered while interpreting J values. For instance, a high variability in the J values was observed for intermediates, especially for the instrument held in the dominant hand. However, in dimensionless motion jerk (DJ), we did not observe a high variability among intermediates since higher motion jerk values of the intermediate group were associated with lower task duration (T) and higher path length (PL) values. Therefore, after normalization (multiplication by $T^{5}.PL^{-2}$ factor) to derive DJ from J, the variability was reduced. Furthermore, significantly longer task durations and path lengths observed in novices compared to experts [10,12,16] could undermine the accuracy of J as a metric for motion smoothness assessment. Hence, the assessment of motion smoothness using mean tooltip motion jerk (J) which is not normalized with respect to task duration and path length can lead to erroneous results, especially if task durations and path lengths differ significantly across the skill levels.

Logarithmic dimensionless tooltip motion jerk (DJ) used tooltip motion jerk and normalized it with respect to path length and task duration, transforming it to a dimensionless metric. As opposed to J which failed to show smoother tooltip movements in the experts’ group, experts had lower DJ values than novices and intermediates, indicating smoother tooltip movements. Observation of significant differences in DJ values among skill levels is in accordance with [24]. However, using the DJ metric, Oropesa et al. [10] could not find significantly different values of DJ across expert surgeons, surgical residents, and novices. The reason underlying not observing significant differences might be originated from the formula used by them. They used the magnitude of the position vector rather than the position vector, which was used in this study, to calculate DJ. Using the magnitude of the position vector to calculate the motion jerk could affect the motion jerk as using the magnitude eliminates the differences resulting from the sign changes. Our analysis revealed significant differences in the tooltip path length and task duration across the surgical skill levels (Figure 4 and Table 2). This can explain why the outcome for comparison among the three skill levels based on J and DJ were not similar due to the sensitivity of the former to task duration and path length.

Originally, 95% tooltip motion frequency (f_95%_) was proposed in this study to characterize tooltip motion smoothness using the frequency spectrum of the tooltip position. f_95%_ attributed lower values to smoother tooltip movements since jerky movements occur in the high-frequency ranges of the tooltip position time series. The other two previous motion smoothness metrics, i.e., J and DJ, were obtained from the third time derivative of the tooltip position, which were affected by differentiation noise. However, our proposed metric (f_95%_) did not require a third time derivative of tooltip position calculation; therefore, it was not affected by differentiation noise. Furthermore, it did not require normalization, eliminating the need for path length calculation. f_95%_ differed significantly across different surgical skill levels, indicating its value for motion smoothness assessment. f_95%_ showed a lower value for experts, representing a smoother tooltip movement than intermediates and novices. This was in accordance with the results of DJ.

In this study, tooltip path length and task duration differed significantly across the surgical skill levels, attributing shorter tooltip path lengths and task durations to higher skill levels (Figure 4 and Table 2). Experts were able to reduce unnecessary surgical instrument movements, which led them to move tooltips into shorter trajectories and, in turn, shorter task durations (Figure 5, Figure 6 and Figure 7). This finding was aligned with previous studies, which showed that experts were able to accomplish the tasks with shorter instrument path lengths and in shorter durations [10,12,26,27]. This observation was due to the fact that higher surgical skill levels benefit from better depth perception and eye–hand coordination skills. Owing to the fulcrum effect, causing the opposite direction movements of tooltips and hands, and 2D surgical site view, depth perception, and eye–hand coordination play a vital role in acquiring minimally invasive surgery skills.

This study had some limitations. We included relatively simple tasks in this simulation study as we intended to recruit novice participants. The implemented tasks were not actual surgical tasks as we did not intend to hurt patients during data collection. Cautions will be needed for the generalization of our findings to real and advanced laparoscopic procedures. Furthermore, a relatively small number of expert surgeons and surgical residents were recruited in this study. To draw a generalized conclusion based on the findings of this study, larger groups of expert surgeons and surgical residents should be recruited in future studies.

## 5. Conclusions

In this study, we evaluated three motion smoothness metrics to identify the responsive metrics for surgical skill assessment. Our findings showed that motion smoothness metrics derived based on tooltip motion jerk without normalization, such as mean tooltip motion jerk, are not responsive metrics for motion smoothness assessment. On the other hand, dimensionless tooltip motion jerk and 95% tooltip motion frequency (originally proposed in this study) were able to reveal tooltip motion smoothness differences among surgical skill levels. These two metrics indicated that higher surgical skill levels were associated with smoother tooltip movements. Hence, we recommend these two metrics for objective surgical skill assessments, which can, in turn, lower the damage to body organs, blood vessels, and nerves due to jerky surgical instrument movements or hand tremors. In future studies, these motion smoothness metrics should be implemented for each subtask, in addition to the entire task. In addition, larger groups of experts and intermediates should be recruited along with more complex surgical tasks to further characterize the impact of surgical skill level on the tooltip motion smoothness.

## Figures and Tables

**Figure 1 sensors-23-03146-f001:**
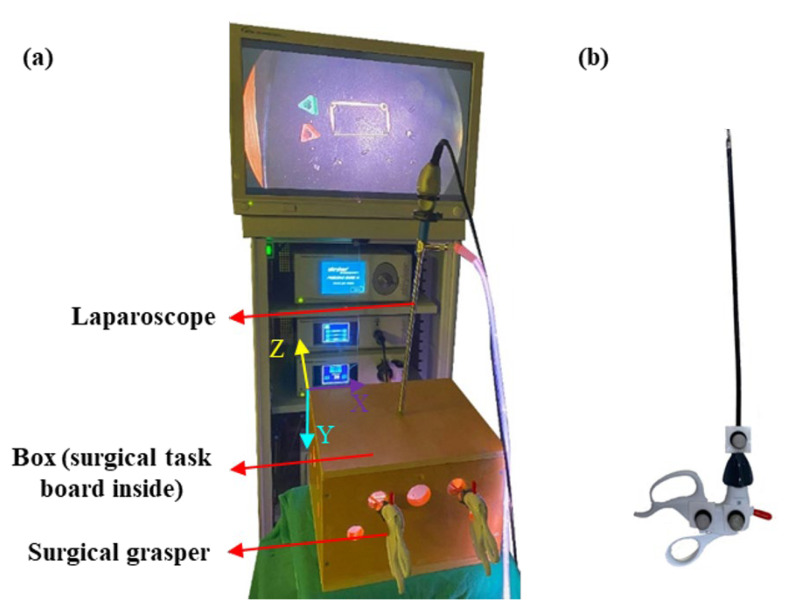
(**a**) Simulated laparoscopic surgical setting, including a box, surgical graspers, a laparoscope, and a Stryker laparoscopic surgery setup, and (**b**) a surgical grasper with markers placed on its handle.

**Figure 2 sensors-23-03146-f002:**
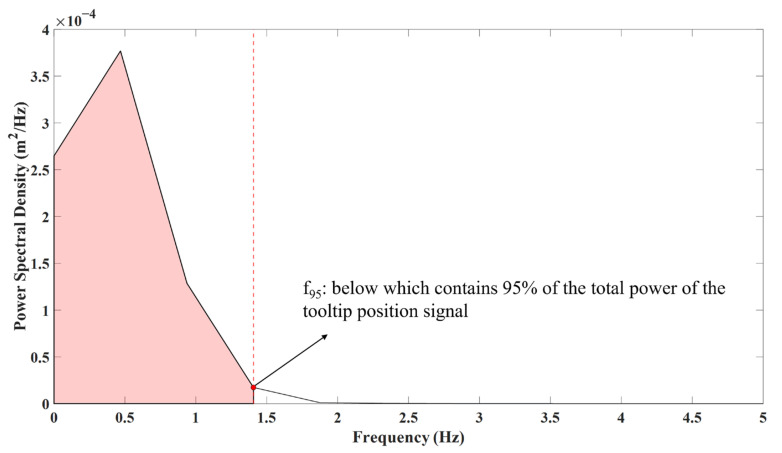
The power spectral density of a representative tooltip position signal. f_95%_ was calculated as the frequency below which contains 95% of the total power of the tooltip position signal, indicated by the red dashed line. The red-shaded area accounts for 95% of the total power of the tooltip position signal.

**Figure 3 sensors-23-03146-f003:**
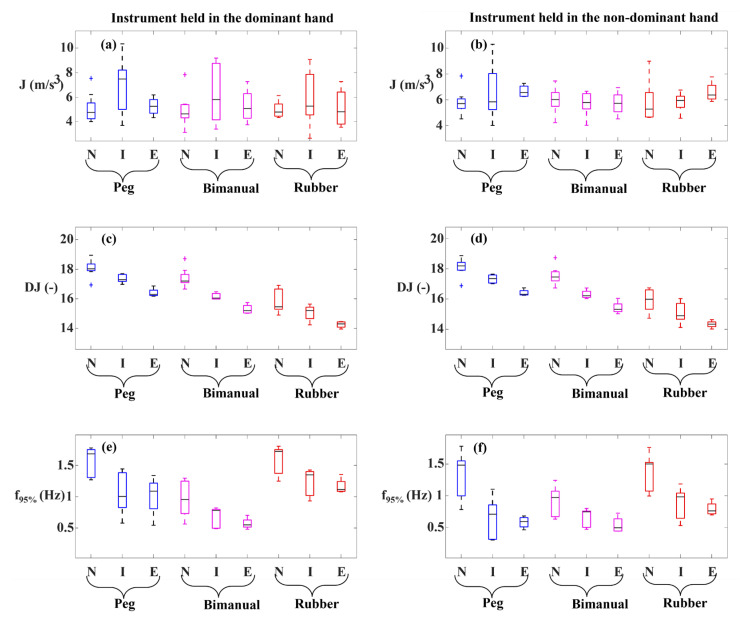
Tooltip motion smoothness metrics in the three tasks: Peg transfer, Bimanual peg transfer, and Rubber band translocation. (**a**) Mean tooltip motion jerk (J) of the instrument held in the dominant hand, (**b**) mean tooltip motion jerk (J) of the instrument held in the non-dominant hand, (**c**) logarithmic dimensionless tooltip motion jerk (DJ) of the instrument held in the dominant hand, (**d**) logarithmic dimensionless tooltip motion jerk (DJ) of the instrument held in the non-dominant hand, (**e**) 95% tooltip motion frequency (f_95%_) of the instrument held in the dominant hand, and (**f**) 95% tooltip motion frequency (f_95%_) of the instrument held in the non-dominant hand. N, I, and E indicate Novice, Intermediate, and Expert groups, respectively.

**Figure 4 sensors-23-03146-f004:**
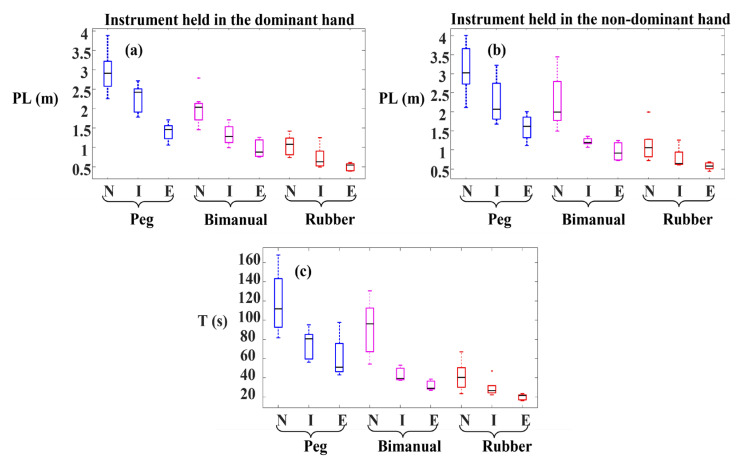
Tooltip path length (PL) and task duration (T) in the three tasks: Peg transfer, Bimanual peg transfer, and Rubber band translocation. (**a**) Path length of the instrument held in the dominant hand, (**b**) path length of the instrument held in the non-dominant hand, and (**c**) task duration. N, I, and E indicate Novice, Intermediate, and Expert groups, respectively.

**Figure 5 sensors-23-03146-f005:**
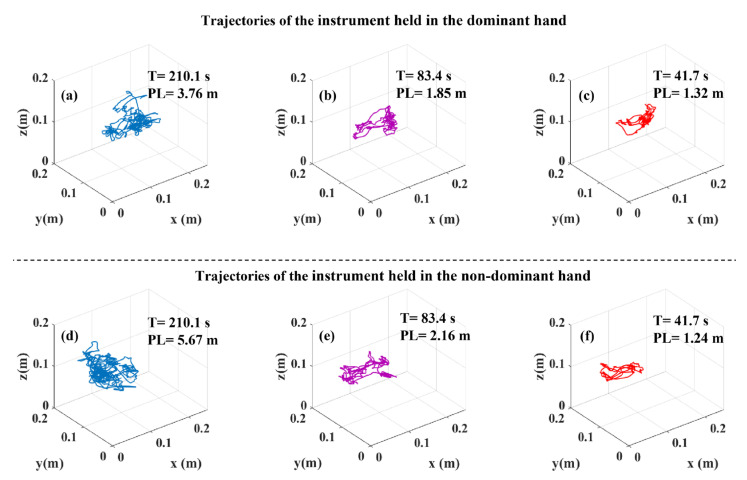
Representatives of tooltip trajectory with tooltip path length (PL) and task duration (T) values in the peg transfer task: (**a**) instrument held in the dominant hand of a novice participant, (**b**) instrument held in the dominant hand of an intermediate participant, (**c**) instrument held in the dominant hand of an expert participant, (**d**) instrument held in the non-dominant hand of a novice participant, (**e**) instrument held in the non-dominant hand of an intermediate participant, and (**f**) instrument held in the non-dominant hand of an expert participant.

**Figure 6 sensors-23-03146-f006:**
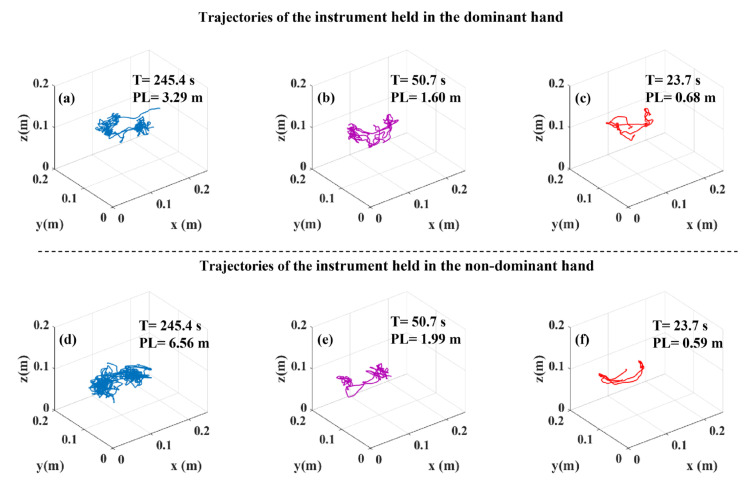
Representatives of tooltip trajectory with tooltip path length (PL) and task duration (T) values in the bimanual peg transfer task: (**a**) instrument held in the dominant hand of a novice participant, (**b**) instrument held in the dominant hand of an intermediate participant, (**c**) instrument held in the dominant hand of an expert participant, (**d**) instrument held in the non-dominant hand of a novice participant, (**e**) instrument held in the non-dominant hand of an intermediate participant, and (**f**) instrument held in the non-dominant hand of an expert participant.

**Figure 7 sensors-23-03146-f007:**
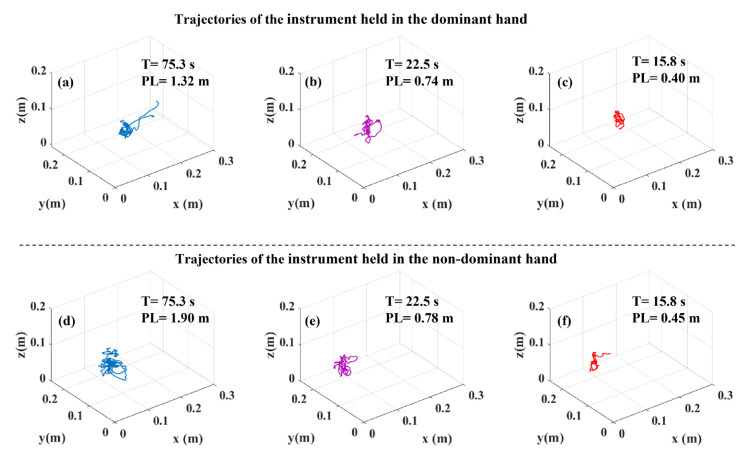
Representatives of tooltip trajectory with tooltip path length (PL) and task duration (T) values in the rubber band translocation task: (**a**) instrument held in the dominant hand of a novice participant, (**b**) instrument held in the dominant hand of an intermediate participant, (**c**) instrument held in the dominant hand of an expert participant, (**d**) instrument held in the non-dominant hand of a novice participant, (**e**) instrument held in the non-dominant hand of an intermediate participant, and (**f**) instrument held in the non-dominant hand of an expert participant.

**Table 1 sensors-23-03146-t001:** Tooltip motion smoothness metrics for surgical skill assessment and their formulas. The tooltip position of the surgical instruments is denoted by $\vec{r}.\;$
PL and T represent tooltip path length and task duration, respectively.

Motion Smoothness Metric	Formula
Mean tooltip motion jerk	$J=\frac{1}{T}\;\int_{t=0}^{T}\left|\frac{d^{3}\vec{r}}{dt^{3}}\right|dt$
Logarithmic dimensionless tooltip motion jerk	$DJ=\ln\left(\frac{T^{5}}{PL^{2}}\int_{t=0}^{T}{\left|\frac{d^{3}\vec{r}}{dt^{3}}\right|}^{2}dt\right)$
95% tooltip motion frequency	$f_{95\%}=$ The frequency below which contains 95% of the total power of the tooltip position signal.

$T=\frac{n}{f_{s}}\;,\;\;n:$ number of motion tracking samples, $f_{s}$: motion tracking sampling frequency, $PL=\int_{t=0}^{T}\left|\frac{d\vec{r}}{dt}\right|dt$.

**Table 2 sensors-23-03146-t002:** Tooltip motion smoothness, tooltip path length, and task duration comparison among the three surgical skill levels. *p*-values associated with the three motion smoothness metrics, path length, and task duration in the three tasks for the instruments held in the dominant and non-dominant hands are presented in the table. *p*-values below the significance level (0.05) are shown in bold font and (*).

Metric	Task	The Instrument Held in the Dominant Hand				The Instrument Held in the Non-Dominant Hand			
		All	N-E	N-I	I-E	All	N-E	N-I	I-E
Mean tooltip motion jerk (J)	Peg transfer	0.580	0.825	0.438	0.413	0.043 *	0.003 *	0.438	0.556
	Bimanual peg transfer	0.736	0.825	0.518	0.730	0.970	0.940	0.797	1.000
	Rubber band translocation	0.698	0.604	0.518	0.730	0.114	0.050	0.438	0.286
Logarithmic dimensionless tooltip motion jerk (DJ)	Peg transfer	**0.002 ***	**0.003 ***	**0.019 ***	**0.016 ***	**0.002 ***	**0.003 ***	**0.019 ***	**0.016 ***
	Bimanual peg transfer	**0.001 ***	**0.003 ***	**0.001 ***	**0.016 ***	**0.001 ***	**0.003 ***	**0.001 ***	**0.032 ***
	Rubber band translocation	**0.006 ***	**0.003 ***	0.060	0.063	**0.011 ***	**0.003 ***	0.147	0.111
95% tooltip motion frequency (f_95%_)	Peg transfer	**0.020 ***	**0.020 ***	**0.029 ***	0.905	**0.004 ***	**0.003 ***	**0.007 ***	0.730
	Bimanual peg transfer	**0.023 ***	**0.011 ***	0.083	0.413	**0.024 ***	**0.020 ***	0.083	0.190
	Rubber band translocation	**0.024 ***	**0.011 ***	0.060	0.905	**0.004 ***	**0.003 ***	**0.007 ***	0.730
Tooltip path length (PL)	Peg transfer	**0.003 ***	**<0.001 ***	0.240	**0.008 ***	**0.005** *****	**0.002 ***	0.083	0.056
	Bimanual peg transfer	**0.002 ***	**<0.001 ***	**0.012 ***	0.095	**<0.001** *****	**<0.001 ***	**<0.001 ***	0.095
	Rubber band translocation	**0.007 ***	**<0.001 ***	0.112	0.222	**0.004** *****	**<0.001 ***	**0.042 ***	0.222
Task duration (T)	Peg transfer	All: **0.001 ***		N-E: **0.001 ***		N-I: **0.007 ***		I-E: **0.016 ***	
	Bimanual peg transfer	All: **0.001 ***		N-E: **0.001 ***		N-I: **0.001 ***		I-E: **0.016 ***	
	Rubber band translocation	All: **0.004 ***		N-E: **0.001 ***		N-I: 0.112		I-E: 0.056	

## Data Availability

Data are available from the authors upon reasonable request.

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
