# Peer review of "Motion Smoothness-Based Assessment of Surgical Expertise: The Importance of Selecting Proper Metrics"

_sensors, 2023, doi:10.3390/s23063146_

Round 1
Reviewer 1 Report
In this research, the authors used a motion capture system to record moving traces of experts, intermediates, and novices in the same simulated surgeries to verify the difference in motion smoothness. Three metrics were adopted to quantify this difference. The use of 95% tooltip motion frequency is reasonable but needs some improvement. I would like to suggest accepting this manuscript if the following questions can be answered appropriately:
1. Instead of using integral of the power spectral density, what are the reasons for the selection of the 95% frequency? How about using RMS phase jitter or phase noise to differentiate the three groups?
2. What is the tracking accuracy of the OptiTrack Flex 13?
3. In authors’ results, dimensionless motion jerk shows clear differences among the three groups. However, in the Introduction, the authors stated controversy in this metric. Is there any way to explain why Ref. 10 could not find that motion smoothness differed significantly?
4. Table 2 is mislabeled.
Reviewer 2 Report
In this work, Aghazadeh et al. develop proper metrics to assess the motion smoothness of surgical performance The overall presentation is generally well structured. The introduction and methods part is clear and informative. However, I do have some concerns about the author's arguments.
1. In figure 3a, could the authors comment on why the intermediate group constantly has the largest error bar for the mean tooltip motion jerk(J) while the instrument was held in the dominant hand?
2. In the discussion, the authors mention: “J would have different values if the same level of motion smoothness was maintained during the trials of the same task with different trial durations, e.g., with different rest periods.” I wonder instead of using the ‘complicated’ logarithmic dimensionless tool motion jerk, why not define a value as J*T? I tried to extract the numbers from the figures and J*T works pretty well.
3. It would help if the authors can derive the DJ equation, especially the (T^5/PL^2) part.
Reviewer 3 Report
There is no established method to express the smoothness of forceps movement using a formula. Evaluation of the smoothness of forceps movement resulted in different results depending on the evaluation methods. Focusing on this fact, the authors tried three algorithms on three tasks. It is of great significance to quantify the smoothness of forceps in the assessment of surgical techniques and in surgical education. Therefore, it is valuable that this paper presented a new evaluation method and was able to discriminate skill level. However, I have some questions.
1. Three motion smoothness derivation algorithms were described. (1) Mean tooltip motion jerk (J), (2) Logarithmic dimensionless tooltip motion jerk (DJ), (3) 95% tooltip motion frequency (f95%). DJ and f95% were admirably differentiating the level of the surgeon. On the other hand, the basic indicators tooltip path length (PL) and task duration (T) seemed to be able to discriminate the skill level of more sensitively. In the first place, is the smoothness of forceps important in assessing surgical techniques? Wouldn't PL or T be sufficient for assessment? The authors should explain the importance of assessment of forceps smoothness. In my opinion, smooth movement contributes to precision surgery (e.g. resection margins) rather than reducing the risk of organ damage.
2. The motion capture system was used to acquire the position data of surgical instruments. Three markers were attached to the handle portion of the forceps. Can the markers be blocked from the camera by the operator's hand?
3. Participants were divided into expert, intermediate, and novice group. The boundary between expert and intermediate group is 50 laparoscopic surgery cases. The number of laparoscopic surgery cases should be described for each group.
4. The authors should describe the small sample size in the limitation section.
Round 2
Reviewer 2 Report
The authors' responses are sufficient, although I still think the largest error bar of motion jerk (J) of the intermediate group results from insufficient data collection. Nevertheless, I would like to recommend the manuscript to publish in its present form.